# HOLISTIC PROMPTING: JOINT REASONING WITH REUSABLE STATES AND SHORTCUT DISCOVERY

## ABSTRACT

Large Language Models (LLMs) have demonstrated significant capabilities in complex reasoning tasks, often employing frameworks like Tree of Thoughts (ToT) and Chain-of-Thought (CoT). However, such methods typically rely on trajectory-based state representations, where each state encapsulates the entire history of reasoning steps. For tasks with inherent solution spaces where overlapping reasoning paths frequently emerge, this inherently restricts the reuse of intermediate computations, leading to redundant exploration. The effect is even more dramatic when samples can be solved jointly, as overlapping reasoning paths frequently occur across different problem instances. We present Holistic Prompting, a novel framework that empowers LLMs to reuse intermediate results both within and across problem instances. Designed for tasks that exhibit reusable sub-structures, Holistic Prompting unifies shared access to intermediate thoughts with an active shortcut-discovery mechanism, enabling focused search between unsolved and solved subproblems and aggressively pruning reasoning paths. Our experiments show that reuse is highly profitable in ToT-style breadth-first search on the math puzzle Game24 and in AlphaZero-style Monte-Carlo tree search in retrosynthetic planning. Here, Holistic Prompting achieves both higher success rates, while at the same time requiring fewer model invocations and outputs.

## 1 INTRODUCTION

Although Large Language Models (LLMs) continue to show remarkable performance over a wide variety of tasks, a challenging problem still lies in the generalist nature of their generation process. Recently, multi-step prompting techniques, such as Chain-of-Thought (CoT) (Wei et al., 2022; Kojima et al., 2022) or Tree of Thoughts (ToT) (Yao et al., 2023a), have been hailed as essential remedy to render the generation process more task-aware by allowing for certain reasoning capabilities. However, this remedy comes at a price: exploring the necessary intermediate steps for each problem instance in detail is expensive in terms of test-time computation. The central problem is that, for each new instance, even of the same problem class, the prompting is re-evaluated, i.e., the LLM starts from scratch every time.

This behavior seems counterintuitive, since many problems, especially those of the same or at least a similar problem class, share a common way of addressing them. However, LLMs cannot readily reuse information learned at different stages during a multi-step prompting. The reason is that such prompting techniques typically rely on trajectory-based state representations, where each state encapsulates the entire history of reasoning steps, to force coherence in subsequent generation steps. On the other hand, we have seen huge success for the Retrieval-Augmented Generation (RAG) paradigm (Lewis et al., 2020), where concise and topically focused knowledge is injected from outside sources into LLM generation. So why should LLMs not benefit from their own progress in related problem instances?

In brief, the central question essentially boils down to opening up the state-space of intermediate results for reuse, while at the same time avoiding complexity problems by the obvious combinatorial explosion of possible states. In this paper, we propose *Holistic Prompting* (HP), a framework specifically designed for tasks that exhibit reusable sub-structures. HP lets LLMs work on all instantiations of a task in an integrated and informed fashion, while employing aggressive pruning techniques to keep the state space's complexity at bay. We show how this change in methodology

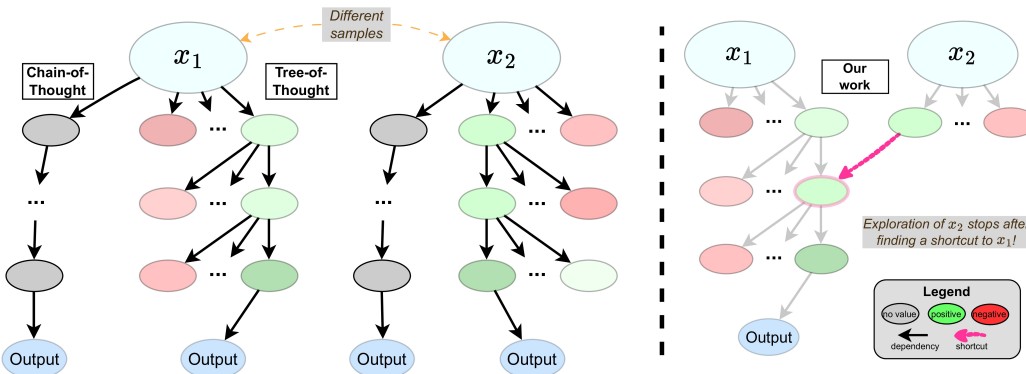

(a) Exemplary CoT and ToT traces on two samples.

(b) Holistic Prompting

Figure 1: High-level overview of CoT, ToT, and Holistic Prompting (ours) and their respective approaches on samples with overlapping intermediates.

can actually be expressed quite naturally by a shift of the underlying state representation. By connecting unsolved problems to previously explored paths of other samples, the LLM can (i) short-cut expensive reasoning traces and therefore save test-time budget, (ii) focus verification efforts on the most frequently traversed reasoning paths to support risk-based resource allocation. Our extensive experiments show that Holistic Prompting does indeed realize substantial gains in success rates while at the same time severely reducing the amount of generated tokens and model calls.

## 2 BACKGROUND

A notable trend in the ever-evolving landscape of prompting techniques has been the incorporation of external thought structures into the generation process. LLMs no longer just produce a single output for a given input (IO-prompting); they are now provided with reasoning scaffolding and are tasked to fill in the missing gaps. The *thoughts* are typically implemented as text spans of varying lengths, ranging from single tokens over paragraphs to full documents. We briefly formalize important baselines in the following. We use $p_\theta$ to represent a pre-trained LLM with parameters $\theta$ that assigns a probability to a token sequence $x = (x^{(1)}, x^{(2)}, \ldots, x^{(N)})$ via the joint probability: $p_\theta(x) = \prod_{i=1}^{N} p_\theta(x^{(i)} \mid x^{(<i)})$. **IO-prompting** directly generates an output $y$ by sampling from $p_\theta$ conditioned on an input $x$, typically framed within task-specific instructions $\text{prompt}_{\text{IO}}$. We write $y \sim p_\theta(\cdot \mid \text{prompt}_{\text{IO}}(x))$, or simply $y \sim p_\theta^{\text{IO}}(\cdot \mid x)$ as a shorthand notation. **Chain-of-Thought** (CoT) (Wei et al., 2022; Kojima et al., 2022) augments the basic generation process by introducing a sequence of intermediate reasoning steps $m_1, \ldots, m_n$, expressed in natural language, to help break down complex tasks into smaller, more manageable parts. This decomposition allows models to handle problems requiring multi-step reasoning. The LLM is used to generate a sequence of thoughts $z_1, \ldots, z_n$: At each step, the model generates the next thought $z_i$ based on the input $x$, all previous outputs, and the associated reasoning steps. Specifically, we sample $z_i \sim p_\theta^{\text{CoT}}(\cdot \mid x \parallel m_1 \parallel z_1 \parallel m_2 \parallel \cdots \parallel m_i)$ where each $m_i$ acts as a thought-provoking cue guiding the next generation. The **Tree of Thoughts** (ToT) (Yao et al., 2023a) framework proposes exploring multiple reasoning paths simultaneously while pruning unpromising ones early. ToT extends the linear chain structure of CoT into a searchable tree of different approaches to a problem instance. Each node in the tree encodes a partial solution to the sample $x$ as a trace of thoughts. Nodes, also called states, are defined as $s^{\text{ToT}} = (x, z_{1\ldots i})$. A generic ToT setup then requires three distinct procedures, each invoked sequentially to: `select` explore-worthy states, `expand` the selected states with possible next sub-states, and `evaluate` them by estimating their utility towards solving $x$. Different search strategies can be employed to decide the exact behavior of each component. Therefore, a full formalization is only possible by choosing a search strategy a priori. Algorithm 1 in the Appendix outlines a breadth-first search as proposed by Yao et al. (2023a).

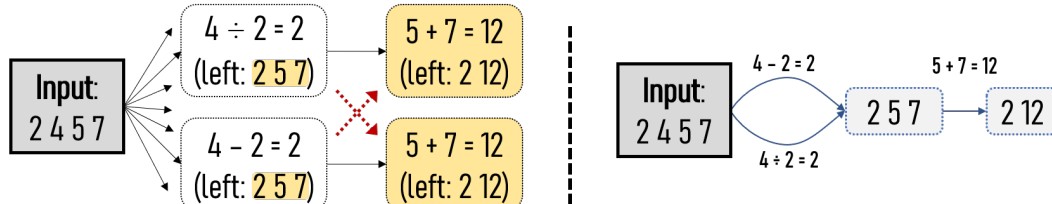

(a) Trajectory-based states (ToT, CoT, CoT-SC., ...)        (b) Collapsed states (ours)

Figure 2: Two excerpts of attempts at Game24 using different state encodings. Redundancies are highlighted in yellow color, ✕ indicates unnecessary branching.

## 3    TOWARDS REUSABLE STATES

We begin with a closer examination of the concept of *state* within previous works to illustrate precisely how current approaches actively restrict the reuse of intermediate results. To demonstrate this, we focus on the ToT framework, as Yao et al. (2023a) have already established it as a generalization of many previous methods, including but not limited to IO-prompting, Chain-of-Thought, and Self-Consistency (Wang et al., 2023). Here, each state is formally defined as $s^{\text{ToT}} = (x, z_{1:i})$, where $x$ denotes the initial input and $z_{1:i}$ represents the sequence of thoughts generated up to step $i$. Thus, each state $s$ encodes the complete reasoning trajectory, from the initial input through the sequence of intermediate steps. Such a setup already suffers from redundancy and branching even when applied to single samples, as depicted in Figure 2a. The reuse between multiple samples, which we consider in this work, is further complicated, since the samples themselves are part of their associated states. Therefore, it is not straightforward to match states of different samples. Moreover, transformations and their outcomes are strongly tied, which unnecessarily inflates the number of unique states.

**Reusable states** To enable reuse of intermediate results, we propose a state encoding that naturally merges equivalent subproblems. For this, we require states to be *self-contained* and *Markovian*, i.e. allowable actions and successors depend only on the current state, not the trajectory taken to reach it. This representation is particularly applicable to problems exhibiting *optimal substructure*, where optimal solutions can be built from optimal solutions to subproblems. Formally, a state $s \in \mathcal{S}$ is decomposed by action $a \in \mathcal{A}$ into a set of sets of subproblems $\hat{S} \subseteq \mathcal{P}_+(S)$. We understand $\hat{S}$ to be a disjunctive decomposition of $s$, meaning that solving any single of the alternatives $\hat{S}_i$ suffices to solve $s$. The $\hat{S}_i$ in turn are conjunctive decompositions, which require all their $\hat{s}_{ij} \in \hat{S}_i$ to be solved for an $\hat{S}_i$ to count as solved. A state with no applicable actions ($\mathcal{A}(s) = \emptyset$) is terminal (irreducible) and is directly assigned a Boolean constant as its solved status.

Choosing nodes labeled by states and edges labeled by actions then gives an And-Or graph structure (for an overview of classical And-Or trees, see Khorasani, 2008). Together with our proposed state representation, this construction supports sharing and reusing partial solutions within a sample, as multiple action sequences may converge to the same state (see Figures 1b and 2b). Reuse *across* problem instances follows naturally by allowing multiple samples to live in the same graph.

This structure subsumes and generalizes two important lines of work on LLM problem decomposition, specifically ToT and Decomp (Khot et al., 2023). Decomp restricts every state to a single `And` node ($|\hat{S}| = 1$). ToT, in contrast, permits only `Or` nodes of size one ($|\hat{S}_i| = 1$). In both cases each sample is processed independently (one instance per graph) and each sub-problem is solved in isolation, so the induced search structures degenerate to trees rather than general graphs.

We define our And-Or graph as a directed hypergraph $G = (V, E)$ where $V$ is a non-empty set of nodes with hyperedges $E \subseteq V \times \mathcal{P}_+(V)$. In a typical And-Or tree, whether a node is solved can be determined by backing up from terminals and iteratively updating their single parent. Since our And-Or *graph* is no longer guaranteed to be acyclic and allows nodes to have multiple parents, the question arises whether this procedure is still valid or even terminates. In A.2 we show that synchronous iteration over the intermediate states in the graph always converges to a least fixed point, due to the monotonicity of `And`/`Or`.

# 4 HOLISTIC PROMPTING

We introduce *Holistic Prompting*, a search-based framework that enables LLMs to reason over multiple samples jointly. Unlike conventional approaches that treat each sample independently, our method extends the search space to include all input problems simultaneously, allowing intermediate computations and solutions to be shared across examples. Central to our approach is a formal representation of a reasoning graph over *reusable states*: nodes correspond to subproblems, and edges denote actions that transform some aspect of their originating state, as outlined in the previous section. Inspired by classical search algorithms, an iteration in our framework involves four core functions: (1) proposing promising actions based on current states across *any* of the problem instances; (2) executing these actions to derive subproblems; (3) assessing whether the proposed actions and new states are legal, and (4) estimating the utility of these new states towards their respective final solutions.

Our approach enables the development of global solution strategies, including the novel shortcut discovery method introduced in this paper. This can be understood as a form of inverse search, where the system actively seeks actions that connect pre-existing, known states, thereby deliberately attempting to generate reuse. In the following, we will delve deeper into the four core functions (1-4) and discuss how they are operationalized within our framework. Section A.3 details Holistic Prompting in the style of breadth-first search (HP-BFS) and MCTS (HP-MCGS).

**(1) Action Generation** Given a non-empty reasoning graph $G = (V, E)$ where $X \subseteq V$, we sample action candidates $\hat{a}$ for state $v \in V$ using two complementary methods.

(1a) *Unconstrained action generation* refers to searching for generally applicable actions on some state $v$. The search is usually broad, but possibly non-exhaustive, due to the probabilistic nature of some generators, especially LLMs. Therefore, some permissible actions may never be found. Standard prompting schemes, e.g. IO or CoT, can be utilized by sampling from the an LLM directly. For example, a set of action candidates $\{\hat{a}_1, \hat{a}_2, \dots\}$ with $\hat{a}_i \sim p_\theta^{\text{propose}}(\cdot \mid v)$ could be obtained.

(1b) *Shortcut discovery* refers to a new type of action generation, aiming to uncover missing links between known states. The motivation is clear: although the aforementioned action generation may eventually discover some of the shared subproblems, the process is mainly driven by chance and may miss. Here, we target a mechanism that directly searches for *useful* action candidates, where the exact semantics of usefulness can be adapted to fit the desired search behavior. For example, problem classes that frequently share subproblems at shallow layers may benefit from initially constructing large connected components in $G$, allowing later computation to focus on critical paths. Formally, given two states $v, v' \in V$, we are searching actions $a \in \mathcal{A}$ such that $(v, v', a)$ is a legal transition. We experiment with a simple but general approach that utilizes an LLM's inverse planning capabilities. For a shortcut candidate $(v, v')$, we infer action candidates via $ac \sim p_\theta^{\text{shortcut}}(\cdot \mid v, v')$. We leave the development of more elaborate discovery mechanisms, e.g., pattern-based or analogous reasoning, to future work. To account for large search spaces, we introduce a candidate evaluator $c : V \times V \to \mathbb{R}$ and run only the top $k$ queries.

**(2) Next State Generation** Having generated action candidates, the next step is to apply them, thereby producing new states. Here, it is critical that the actions are carried out correctly, since the reuse of illegal states can affect more than a single sample in our framework. Given an action candidate $\hat{a}$ for some state $v$, we sample the next states as $\hat{V} \sim p_\theta^{\text{apply}}(\cdot \mid \hat{a}, v)$.

**(3) Action and State Verification** A major concern for any iterated reasoning method with imperfect generators is the accumulation of errors. Especially undetected errors in shallow layers can lead to whole subgraphs being invalidated. While this is a general challenge for more complex reasoning structures, it becomes particularly acute in our framework, where intermediate results are explicitly designed for reuse across different samples. Depending on the exact problem at hand, verification may be implemented as external feedback to the procedure. This requires explicating the rules and regulations as programs, which might not be feasible for all tasks. In our exemplary implementation, we instead take inspiration from recent works on the ability of LLMs to self-verify (Li et al., 2023; Madaan et al., 2023). Thus, we leverage the LLM to verify that found actions and derived states are indeed correct. Dedicated instructions are used, demanding a binary decision about action or state legality, with few-shot examples of common errors to guide the decisions. We introduce two sets of prompts $\mathcal{P}_{\text{action}}$ and $\mathcal{P}_{\text{state}}$ in the form of yes/no queries. Each prompt is considered to encode a rule

of the task, so that action and state candidates can be subjected to rigorous verification. To account for variance, we may sample $m$ times and take the average.

**(4) State Evaluation** After generating and verifying new states, the framework must prioritize which states to expand next in the reasoning graph. This is accomplished through *state evaluation*, which estimates the utility of each candidate state w.r.t. the solution process and guides the search strategy. To this end, we introduce a *state evaluator* $r(G, \hat{v})$, which scores each candidate node based on the entire structure of the current reasoning graph $G$. Unlike previous approaches (Huang et al., 2023; 2025; Yao et al., 2023a) that typically rely on local information for node evaluation, our method enables global state assessments. In particular, $r(G, \hat{v})$ may reflect various heuristics, such as the estimated cost to reach a final solution from $\hat{v}$, the likelihood that $\hat{v}$ is part of an optimal solution, or the potential of $\hat{v}$ to enable future reuse across multiple problem instances. By guiding expansion using these global evaluations, the framework can more efficiently explore the reasoning space and improve overall search performance.

In summary, our framework generalizes typical search paradigms (expand, evaluate, select) to the multi-sample setting, enabling solution sharing, self-correction, and global solution strategies, here studied in the form of shortcut discovery.

## 5 EXPERIMENTS

To thoroughly analyze our method's potential we conduct experiments over a theoretical math puzzle setting and a real world chemical retrosynthesis task. Both settings require compositional search and planning: in the Game24 math puzzle, the goal is to reach a target sum of 24 given a fixed set of four numbers and the basic arithmetic operators. In retrosynthetic planning, a target molecule must be recursively decomposed into precursor molecules until reaching commercially available building blocks. Both tasks pose important, yet orthogonal challenges for search methods, since math puzzles may be unsolvable and chemical reaction graphs are often highly cyclic. We implement our framework using breadth-first search (HP-BFS) and the PUCT variant of MCTS, as popularized by AlphaZero (HP-MCGS).

### 5.1 GAME24

In Game24, a player is given four numbers (e.g., "2 4 5 7") and the basic arithmetic operators $(+, -, \times, \div)$ and has to form an expression that evaluates to 24, using each number exactly once. This well-defined setting is often used to assess the reasoning capabilities of LLMs (Sel et al., 2024; Yao et al., 2023a). For analysis we differentiate two scenarios. First, in a constructive setting, we measure how well problems known to be solvable are tackled, which is common in related work (Besta et al., 2024; Khot et al., 2023; Zhou et al., 2023). Second, in a decisional setting, we ask whether a given instance is solvable. As a twist, none of our test cases in this setting are solvable, precluding proofs by construction from a single found solution and instead requiring the considerably harder evidence of absence. Since this demands more thorough search it makes the task particularly suited for stress-testing compositional reasoning.

**Datasets.** We use two 100-sample datasets. For the constructive setting, we adopt the ToT test split (Yao et al., 2023a) (ToT-Test-Split). For the decisional setting, we introduce Unsolvable-100, for which no combination yields 24. Both datasets are listed in the appendix in Section A.5, accompanied by further dataset statistics, including state and action counts under different state representations.

**LLM Setup.** We use MistralAI's Mistral-Small-3 (24B), Microsoft's Phi-4 (14B) and Meta's Llama-3.3 (70B) to account for differently sized LLMs in our experiments. Prompts are run with a chat completion setup, sampling with a temperature of $0.7$, with the exception of yes/no queries, where we sharpen the distributions further with a temperature of $0.1$.

**Metrics.** We report success rate as the proportion of correctly answered samples. For solvable samples, correctness means giving any correct solution, whereas for unsolvable samples, the LLMs must not provide any solution or claim to have found one. It is important to note that the correctness notion of success is different from the classical understanding for decision problems, where always a complete logical proof is required. While LLMs may enumerate all possible approaches to solve the problem at hand and thus also provide constructive proof by exhaustion in case there is no solution,

Table 1: Best results of various prompting methodologies on two datasets of Game24. Exact hyperparameters are shown in Section A.5.

| Metric | Phi-4 (14B) | | | | Mistral-Small-3 (24B) | | | | Llama-3.3 (70B) | | | |
|---|---|---|---|---|---|---|---|---|---|---|---|---|
| | IO$_{SC}$ | CoT$_{SC}$ | ToT$_+$ | HP | IO$_{SC}$ | CoT$_{SC}$ | ToT$_+$ | HP | IO$_{SC}$ | CoT$_{SC}$ | ToT$_+$ | HP |
| **Dataset: ToT-Test-Split** | | | | | | | | | | | | |
| %Success | 0.19 | 0.15 | 0.78 | 0.96 | 0.34 | 0.10 | 0.626 | 0.87 | 0.51 | 0.25 | 0.96 | 0.98 |
| #M Tokens | 3 | 3 | 119 | 13.1 | 7.4 | 6.6 | 71.9 | 12.7 | 6.3 | 5.9 | 75.3 | 10.6 |
| #K Calls | 20 | 20 | 617 | 117 | 20 | 20 | 424 | 119 | 20 | 20 | 606 | 141 |
| **Dataset: Unsolvable-100** | | | | | | | | | | | | |
| %Success | 0.01 | 0.0 | 0.82 | 0.998 | 0.27 | 0.92 | 0.74 | 1.0 | 0.73 | 0.52 | 0.786 | 1.0 |
| #M Tokens | 4.7 | 4.9 | 111 | 21.7 | 5.6 | 8.2 | 66 | 12.6 | 7.7 | 8.7 | 70.6 | 9.6 |
| #K Calls | 20 | 20 | 571 | 189 | 20 | 20 | 359 | 137 | 20 | 20 | 563 | 133 |

we do not enforce it here. Rather, we regard unsolvable instances as successful, in which an LLM has not found a solution after thorough search. As additional metrics for comparing models, we collect the sum of completion tokens and the number of calls to the model.

**Baselines.** We use IO and CoT baselines with a self-consistency extension over 100 tries (Wang et al., 2023). We include ToT-BFS using the official implementation.[1] To further strengthen ToT and for a fair comparison, we implement the same self-verification as in our own framework. Hyperparameter search over breadth $b \in \{1, 3, 5, 7, 10, 20\}$, the number of evaluations $m \in \{3, 7, 11\}$, and whether to use verification is conducted to find well-working setups for maximum success rates. If enabled, verification is averaged over three prompts. ToT is run in an oracle setup (ToT$_+$), where we count an output as correct if there is any correct answer within the top $1 \ldots b$ predictions. Wherever possible, we use the same prompts for baselines and our framework, and ensure that all setups have access to exactly the same few-shot examples and rules.

**Holistic Prompting Setup.** To apply HP to Game24, we decompose each instance into a sequence of subproblems: In each step, two numbers are combined into a new one, ideally converging to a single result. An overview is shown in Figure 3 in the Appendix. We explore a combination of local and global evaluation to inform the selection of the next states. Locally, the LLM is utilized to estimate $\hat{v}$'s likelihood of leading to a solution. We incorporate global information by assigning states a value of 0 unless they are reachable from at least one currently unsolved sample:

$$r(G, \hat{v}) \sim p_\theta^{\text{value}}(\cdot \mid \hat{v}) \times \mathbf{1}[\exists x \in X : x \overset{G}{\leadsto} \hat{v}].$$

Throughout all experiments, we choose a simple heuristic to implement the shortcut candidate evaluator $c$ based on the Levenshtein string edit distance. The rationale behind it is the necessary overlap between the string representations of a parent and its legal child nodes. While this helps to identify true negatives, it is bound to suffer from many false positives. For example, the edit distance from "2 3 11" to "5 11" is equal to that of "2 3 11" to "7 11", although no legal move exists that produces the latter. We choose this suboptimal component deliberately to show that even weak signals suffice to uncover helpful shortcuts. Other methods, e.g. learned classifiers or distance measures exploiting task semantics, are left to future work. Regarding hyperparameters, we keep the number of evaluations per state fixed at $m = 3$, explore $b = 10$ states per iteration, and run verification three times. We apply a small search over the number of shortcut candidates for $k \in \{10, 50, 100, 200\}$.

**Results.** As shown in Table 1, ToT+ performs best among the baselines in terms of success rate on the ToT-Test-Split. However, these improvements come at the cost of vastly higher completion token usage and calls to the model. Yet, HP even surpasses ToT+ in all metrics, solving significantly more samples with fewer output tokens (reduced by factors of 9.1, 5.6, and 7.1) and less LLM invocations (reduced by factors of 5.3, 3.5, and 4.3). Please note that this occurs even though ToT+ has an unfair advantage due to its oracle access.

The second experiment involves exclusively unsolvable samples, on which we re-apply the model configurations that performed best in the previous task. This is necessary to avoid setups with little search activity, since in this special experiment never providing any solutions would be a guaranteed

---

[1]Official ToT repository (MIT License): `https://github.com/princeton-nlp/tree-of-thought-llm`

way of answering correctly. We test the different methods configured for exhaustive search, as determined on the Tot-Test-Split dataset. With the exception of Phi-4, which almost always claims to have found a valid solution, IO+CoT appear to correctly address more samples here. This, however, is mainly due to their self-consistency extension, which returns the majority answer based on exact match. Concerning ToT, all models suffer approximately the same number of false positive claims, ranging from 18 to 26 percent. In stark contrast, we find just a single claimed solution from any model using the HP framework.

**Error Analysis.** The detailed error analysis in Section A.6 gives insight into the error types that occur during HP and offer a possible explanation for Mistral-Small-3's comparatively low improvements as compared to Phi-4's. Although the former LLM still enjoys significant improvements over the baselines, Phi-4 beats it in terms of success rate, which is surprising given its smaller size. We found Mistral's verification component to make three to six times as many false positive errors, i.e., it considered actually correct states and actions as incorrect and rejected them. This led to the LLM blocking its own path.

**Ablation Study.** Table 2 shows the effects of the verification and shortcut components. Surprisingly, running verification decreases the token usage and model calls, although clearly more work per state is carried out. However, these costs are amortized since erroneous nodes are never further explored and far fewer states need to be processed by the rest of the framework. Shortcuts appear generally beneficial, cutting token usage and improving success rates. When active together, the highest success rate is achieved while using the fewest tokens and model calls.

Table 2: Ablation study with Phi-4 on ToT-Test-Split.

| Verify | Shortcut | #M Tokens | #K Calls | %Success |
|--------|----------|-----------|----------|----------|
| ✗ | ✗ | 28.7M | 221.6K | 52.4 |
| ✗ | ✓ | 15.4M | 122.5K | 91.6 |
| ✓ | ✗ | 20.8M | 201.5K | 94.4 |
| ✓ | ✓ | 13.1M | 116.8K | 96.2 |

## 5.2 RETROSYNTHESIS

Retrosynthetic planning is a real world strategy of designing a synthesis by working backward from some target molecule to simpler, readily available precursors. This involves several tasks: identifying strategic bond disconnections, representing fragments as idealized synthons, mapping them to real reagents and reactions, and iterating until a feasible forward route is obtained. This iterated decomposition can be understood in the sense of a proof tree and directly maps to our understanding of states and actions, here represented by molecules and reactions, respectively. A typical search algorithm cycles between selecting an unexplored molecule, expanding it by applying suitable reactions taken from a library of templates, and checking whether the target is now fully decomposed into basic building blocks. In the literature, such setups always require a test set of target molecules, a library of template reactions, and a set of basic building blocks. Since the latter two are usually fixed across individual targets, reusing previously discovered molecules across different targets can save search costs, given that overlapping routes do exist. However, many previous works tackle targets purely in isolation, often using tree-search heuristics with learned neural priors, c.f. (Segler et al., 2018; Chen et al., 2020; Hong et al., 2023; Zhao et al., 2024a). A notable development towards reuse is given in Xie et al. (2022) introducing a graph neural network to guide the search.

**Holistic Prompting Setup.** Since the retrosynthesis evaluation will need to work in real world settings, an important precondition concerns the ability of current LLMs to actually generate correct chemical molecules and reactions. Unfortunately, this ability proved to be extremely weak. In fact, recent results in Guo et al. (2023) report that on single-step disconnections GPT-4's error rate is at least 88.6%, even with extra scaffolding and few-shot examples. Since planning is usually only necessary for multi-step reactions, often necessitating hundreds of consecutive disconnections, an error rate this high immediately renders the application of off-the-shelf LLMs infeasible. Moreover, put to the test neither the general purpose models used in our Game24 setting, nor current LLMs specifically trained on chemistry tasks (Xia et al., 2025; Narayanan et al., 2025), were able to solve even a single sample of our test dataset.

Thus, for the time being–although we anticipate future LLMs to better fit this scenario–we stuck with non-LLM neural architectures to implement both, action generation and state evaluation. In

Table 3: Summary of the results of various retrosynthetic planners. We report success rates over six iteration budgets, the average number of iterations, as well as the mean reaction and molecule node counts. The latter three correspond to the search budget with 500 iterations.

| Algorithm | Success rate [%] ↑ | | | | | | #Iterations ↓ | # Reaction Nodes ↓ | # Molecule Nodes ↓ |
|---|---|---|---|---|---|---|---|---|---|
| | 50 | 100 | 200 | 300 | 400 | 500 | | | |
| Greedy DFS | - | 38.42 | 40.53 | 44.21 | 45.26 | 46.84 | 300.56 | - | - |
| Retro* | 40.00 | 55.79 | 70.53 | 76.84 | 82.11 | 85.79 | 158.81 | 2632.84 | 3685.31 |
| Retro*+-0 | 56.84 | 67.37 | 83.16 | 92.11 | 94.74 | 96.32 | 97.95 | 1444.52 | 2139.3 |
| EG-MCTS | 81.58 | 87.89 | 91.05 | 94.21 | 94.21 | 94.21 | 55.43 | 828.78 | 1091.65 |
| RetroGraph | 69.47 | 88.42 | **97.89** | 98.95 | **99.47** | **99.47** | 45.13 | 674.23 | 500.44 |
| DreamRetro | - | 93.16 | 96.32 | 97.37 | 97.89 | 98.42 | - | 574.42 | 480.12 |
| PDVN | 93.16 | **96.84** | **97.89** | **99.47** | **99.47** | **99.47** | 20.24 | 486.87 | 417.54 |
| **HP-MCGS** | **93.68** | 95.79 | 96.84 | **98.95** | **99.47** | **99.47** | **19.12** | **317.01** | **246.55** |

particular, we use a proven single-step model from Chen et al. (2020) as action generator. For the state evaluation, we draw on the works of Segler et al. (2018) and Hong et al. (2023) (EG-MCTS), who show that MCTS paired with well-aligned policy and value networks not only finds more synthesis routes, but does so in fewer iterations than competitive baselines. From the latter, we therefore take the two networks to provide state evaluations. This setup has the additional benefit of reducing extraneous influences on the experimental results, since we only intervene on the graph representation and keep everything else exactly as used in the literature.

To combat the inherent redundancy in the tree-structure underlying MCTS, we move to the graph representation as motivated earlier. However, the possible presence of cycles still requires adjustments in the selection and update phases. In classical PUCT-select, cycles can lead to infinite loops, which is why in this work we employ additional backtracking during traversal (see Section A.3 for details). To account for integrated multi-target planning, we also introduce an artificial root node and corresponding actions that allow dynamic selection of each individual target. We refer to the final variant as HP-MCGS.

**Baselines.** To analyze the performance of our approach, we compare it to a broad set of state-of-the-art baselines, in particular EG-MCTS (Hong et al., 2023), Retro* (Chen et al., 2020), Retro*-0 (Kim et al., 2021), RetroGraph (Xie et al., 2022), PDVN (Liu et al., 2023), and DreamRetro (Zhang et al., 2025).

**Metrics.** We quantify the search procedures' effectiveness by the ratio of solved to unsolved targets (success rate). Efficiency is measured through node counts in the final search graphs rather than a direct model call count. This is necessary for a fair comparison, since some baselines opt for multiple small networks, whereas others use a monolithic merged model. Finally, we also report the mean number of iterations per target molecule.

**Dataset.** We follow common practice in the literature and use the same set of commercially available building blocks (approx. 23M) from eMolecules[2] for all approaches. For the test data, we adopt the 190 challenging target molecules (Retro-190) from Chen et al. (2020), as it is widely used in previous work.

**Results.** HP-MCGS achieves state-of-the-art success while being the most efficient planner (see Table 3). It solves 93.68% of targets at 50 iterations (best), 95.79% at 100 (second to PDVN), and matches the top success of 99.47% by 400 iterations, maintaining parity at 500. The search graph size is markedly smaller: 317 reaction nodes and 247 molecule nodes on average, representing roughly 35% and 41% fewer nodes than PDVN, and over 50% fewer than RetroGraph and EG-MCTS. Overall, HP-MCGS delivers SOTA success rates with the fewest iterations and smallest search graphs, validating reuse with multi-target, cycle-aware MCGS as an effective paradigm for retrosynthesis planning.

---

[2] https://downloads.emolecules.com/free/

## 6 RELATED WORK

**Sequential Multi-Step Reasoning** Traditional methods like Chain-of-Thought prompting (Wei et al., 2022; Kojima et al., 2022) elicit reasoning through a step-by-step, sequential process for each sample independently. Self-Consistency (Wang et al., 2023) builds on this by sampling multiple CoTs in parallel and selecting the majority answer. Graph-based methods such as Tree of Thoughts (Yao et al., 2023a) generalize these sequential and parallel reasoning paradigms. Zhou et al. (2023) propose a decomposition approach, dissecting the task into easier subquestions. Chen et al. (2023) and Khot et al. (2023) provide more programmatic perspectives on decomposing problems into sub-tasks, the latter delegating sub-tasks to different specialized LLMs. Unlike our approach, these methods treat each sample independently.

**Search and Planning with LLMs** Planning for the choice of action in accordance with the optimal outcome is a common problem in LLM research (Yao et al., 2023b; Shinn et al., 2023; Huang et al., 2022). The Tree of Thoughts framework (Yao et al., 2023a) introduces structured, tree-based search and planning for complex reasoning tasks, with subsequent works such as Tian et al. (2024) and Zhang et al. (2024) studying self-improvement. Graph of Thoughts (GoT) (Besta et al., 2024) further generalizes the ToT framework by adopting arbitrary directed graphs, enabling the aggregation of multiple reasoning traces into consolidated thoughts. We understand our framework as a topological generalization of ToT and Decomp to And-Or graphs with an emphasis on targeted reuse of previously obtained results across multiple problem instances.

**Self Evaluation and Feedback** LLMs have been shown to be powerful evaluators of their own generations (Kadavath et al., 2022), and are able to refine their outputs through self-feedback (Li et al., 2023; Madaan et al., 2023; Shinn et al., 2023; Huang et al., 2023; 2025; Zhao et al., 2024b). We follow prompting techniques such as ToT (Yao et al., 2023a), Maieutic Prompting (Jung et al., 2022) and Multiagent Debate (Du et al., 2024; Liu et al., 2024) in exploiting such feedback mechanisms for the evaluation of intermediate steps.

**Efficiency Considerations in Reasoning** To alleviate the computational demands of multi-step reasoning, frameworks for improving token efficiency (Xu et al., 2025), parallelizability (Ning et al., 2024), test-time compute scaling (Snell et al., 2024; Muennighoff et al., 2025), and semantic caching (Bang, 2023) have emerged. We pursue an alternative route and aim to improve efficiency by actively promoting reuse over multiple samples.

Other works seek to maintain alignment with previous answers to reduce inconsistency errors (Kassner et al., 2021; Mitchell et al., 2022) or utilize long-term memory to improve future decision making (Shinn et al., 2023). Li & Qiu (2023) and Yu et al. (2024) also consider reuse of thought chains, but solely provided as in-context examples of related reasoning traces. Whereas they perform reuse up front and only then reason, our framework integrates reuse ad hoc as part of the reasoning process.

## 7 CONCLUSION

Structural reasoning approaches can be beneficially enhanced to reuse their own progress in related problem instances by switching to a more suitable state representation. Our Holistic Prompting framework allows to employ global search strategies over all samples of tasks with reusable substructures. We introduce one such strategy in the form of shortcut discovery, a novel technique that actively infers connections between acquired intermediate solutions. We show that reuse and focused verification significantly improve success rates and at the same time reduce resource consumption. While our And-Or graph setup generalizes two important lines of work on problem decomposition, namely Tree-of-Thought and Decomp, our current notion of reuse relies on strict state equality. A promising direction is a lifted abstraction that maps instance-specific states to parameterized templates (via canonicalization, unification, or learned equivalence) while preserving the And–Or structure. Moreover, the Markovian assumption presumes that all relevant information about a subproblem's state is explicitly captured in its representation, which may be challenging in domains with hidden or implicit context. Because our framework actively promotes reuse of previous computations, it can propagate errors and amplify biases. Our condensed reasoning graph, however, could allow bias auditing methods to focus on states of high connectivity, those that, due to their connectedness in the global reasoning graph, have the highest overall impact.

## REPRODUCIBILITY STATEMENT

We accompany our paper with an appendix for supplementary material and the code repository that produced the results reported for our method. In A.2 we provide theoretical proof for the convergence of truth propagation in cyclic And-Or graphs. For our breadth-first variant, 2 gives a brief outline in pseudocode and our code repo provides the implementation we used in `hp_shortcut_rounds.ipynb`. For our PUCT-MCGS, we show pseudocode of the cycle-aware node selection in Algorithm 3 and give the full implementation in `retrosynthesis_hp.ipynb`. In Section A.4, the experimental setup for Game24 is extended with hyperparameter configurations, information regarding compute infrastructure, model identifiers, dataset statistics and error rates.

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

# A   Technical Appendices and Supplementary Material

## A.1   Breadth-First Search in Tree-of-Thoughts

---

**Algorithm 1** ToT-BFS (Yao et al., 2023a)

---

**Require:** Input $x$, LM $p_\theta$,
  step limit $T$, breadth limit $b$
  $\text{expand}(s) \leftarrow$ sample $\mathbf{z} \sim p_\theta^{propose}(\cdot|s)$
  $\text{evaluate}(s) \leftarrow$ sample $v \sim p_\theta^{value}(\cdot|s)$

  $\text{select}(S_t', V_t) \leftarrow \underset{S \subset S_t', |S|=b}{\arg\max} \sum_{s \in S} V_t(s)$

  $S_0 \leftarrow \{x\}$
  **for** $t = 1, \ldots, T$ **do**
    $S_t' \leftarrow \{(s, z) \mid s \in S_{t-1}, z \in \text{expand}(s)\}$
    $V_t \leftarrow (\text{evaluate}(s_i))_{s_i \in S_t'}$
    $S_t \leftarrow \text{select}(S_t', V_t)$
  **end for**

---

## A.2   Proofs

**Truth propagation in cyclic And-Or graphs**   In the following, we use Knaster-Tarski and Kleene fixed point theorems to show that there always exists a least fixed point that can be reached with synchronous iteration. We start with our graph definition as presented in Section 3. We have a graph $G = (V, E)$ with states $V$ and edges $E \subseteq V \times P_+(V)$. We partition $V$ into nonterminal nodes $B$ and terminal nodes, which we further differentiate by unsolved ($T_0$) and solved ($T_1$) status. We count the number of nonterminal nodes as $n = |B|$. For $x \in \{0,1\}^n$, we define its extension $\text{ext}(x) \in \{0,1\}^{|V|}$ by

$$\text{ext}(x)_i = \begin{cases} x_i, & i \in B, \\ 0, & i \in T_0, \\ 1, & i \in T_1 \end{cases}$$

For each $j \in B$, let $g_j$ be the Boolean expression over its inputs/successors in $G$, built using only $\wedge, \vee$ and constants. Further, define

$$F_j(x) = g_j\big(\text{ext}(x)\big), F : \{0,1\}^n \to \{0,1\}^n$$

and let $\leq$ be the coordinatewise partial order on $\{0,1\}^n$: for $x, y \in \{0,1\}^n$ we write $x \leq y$ iff $x_i \leq y_i, \forall i \in B$.

**Lemma 1** (Monotonicity with arbitrary constants)**.** *F is monotone with respect to the coordinatewise order on $\{0,1\}^n$.*

*Proof.* The map $\text{ext}$ is monotone in $x$, and $\wedge, \vee$ are monotone in each argument. Constants in $g_j$ (0 or 1) do not affect monotonicity. Hence $x \leq y \Rightarrow F(x) \leq F(y)$. $\qquad\square$

**Theorem 1** (Synchronous iteration from the bottom converges to a the least fixed point)**.** *Consider the sequence $x^0 = \mathbf{0}$ (the all-zero vector on B) and $x^{k+1} = F(x^k), k \geq 0$. Then:*

1. *The sequence is nondecreasing: $x^0 \leq x^1 \leq x^2 \leq \cdots$.*

2. *It stabilizes in at most $n$ steps: $\exists m \leq n : x^{m+1} = x^m$, $x^m$ is a fixed point of F.*

3. *$x^m$ is the least fixed point of F with respect to $\leq$.*

*Proof.* (1) Since $\mathbf{0} \leq F(\mathbf{0})$, we have $x^0 \leq x^1$. If $x^k \leq x^{k+1}$, then by monotonicity $F(x^k) \leq F(x^{k+1})$, i.e. $x^{k+1} \leq x^{k+2}$.

(2) Since the sequence is non-decreasing, no coordinate ever flips from 1 to 0. Thus, there can be at most $n$ strict increases. So, for some $m \leq n$ we must have $x^{m+1} = x^m$ (fixed point).

(3) Let $y$ be any fixed point. Since $0 \leq y$ and $F$ is monotone, for all $k$ we have $F^k(0) \leq F^k(y) = y$. With $x^k = F^k(0)$, we get $x^k \leq y$ for all $k$, hence $x^m \leq y$. □

**Remark 1** (Arbitrary seeds can oscillate under synchronous iteration). *Although not applicable to our scenario, there exist 2-cycles for arbitrary seeds, even with monotone $F$. E.g. two* Or *gates feeding into each other starting from the seed $(0, 1)$ will swap to $(1, 0)$, then back to $(0, 1)$ and so on.*

## A.3 PSEUDOCODE

---

**Algorithm 2** Holistic Prompting Breadth-First Style Search

---

**Require:** Inputs $X = \{x_1, \ldots, x_n\}$, Action Generator $p_\theta$, step limit $T$, width $b$, State Evaluator $r()$, Shortcut Evaluator $c()$, shortcut limit $k$, prompts $\text{prompt}_{\{\text{propose,shortcut,apply}\}}$, $\mathcal{P}_{\text{action}}$, $\mathcal{P}_{\text{state}}$

$G_0 = (V_0, E_0) \leftarrow (X, \emptyset)$

**for** $t = 1 \ldots T$ **do**

$\quad V_t' \leftarrow \underset{\{V \subseteq V_{t-1} : |V| = b\}}{\text{argmax}} \sum_{v \in V} r(G, v)$ $\qquad \triangleright$ State Evaluation & Selection

$\quad \hat{A}_t \leftarrow \{(v, \hat{a}) \mid v \in V_t', \hat{a} \sim p_\theta^{\text{propose}}(\cdot \mid v)\}$ $\qquad \triangleright$ Action Candidate Generation

$\quad \hat{I}_t \leftarrow \underset{\{V \subseteq V_{t-1}^2 : |V| = k\}}{\text{argmax}} \sum_{(v,v') \in V} c(v, v')$ $\qquad \triangleright$ Shortcut Evaluation & Selection

$\quad \hat{A}_t \leftarrow \hat{A}_t \cup \left\{ (v, \hat{a}) \,\middle|\, \begin{array}{l} (v, v') \in \hat{I}_t, \\ \mathsf{depth}(v) < \mathsf{depth}(v'), \\ \hat{a} \sim p_\theta^{\text{shortcut}}(\cdot \mid v, v') \end{array} \right\}$ $\qquad \triangleright$ Shortcut Candidate Generation

$\quad \hat{A}_t \leftarrow \{(v, \hat{a}) \in \hat{A}_t \mid \nexists\text{pr} \in \mathcal{P}_{\text{action}} : p_\theta^{\text{pr}}(\text{"yes"} \mid v, \hat{a}) < 0.5\}$ $\qquad \triangleright$ Action Verification

$\quad \hat{E}_t \leftarrow \{(v, \hat{V}, \hat{a}) \mid (v, \hat{a}) \in \hat{A}_t, \hat{V} \sim p_\theta^{\text{apply}}(\cdot \mid v, \hat{a})\}$ $\qquad \triangleright$ Next State Generation

$\quad E_t \leftarrow E_{t-1} \cup \{(v, \hat{V}, \hat{a}) \in \hat{E}_t \mid \nexists\text{pr} \in \mathcal{P}_{\text{state}} : p_\theta^{\text{pr}}(\text{"yes"} \mid v, \hat{V}, \hat{a}) < 0.5\}$ $\qquad \triangleright$ State Verification

$\quad V_t \leftarrow V_{t-1} \cup \{\hat{v} \in \hat{V} \mid (v, \hat{V}, \hat{a}) \in E_t\}$ $\qquad \triangleright$ Update Graph

**end for**

---

---

**Algorithm 3** PUCT Selection with Cycle-Avoidance and Backtracking on And-Or Hypergraphs

---

**Require:** Hypergraph $G = (V, E)$ with $E \subseteq V \times \mathcal{P}_+(V)$, root $v_0$

**Require:** Priors $P_{v,u}$, visit counts $N_v, N_{v,u}$, value estimates $Q_{v,u}$, constant $c_{\text{puct}} > 0$

$\text{Out}(v) \triangleq \{u \in V \mid \exists (v, U) \in E : u \in U\}$ $\qquad \triangleright$ one-step successors

$\mathsf{Path} \leftarrow [v_0]$ $\qquad \triangleright$ path stack

$\tau \leftarrow \{v_0\}$ $\qquad \triangleright$ nodes on current trajectory

**while** true **do**

$\quad v \leftarrow \text{top}(\mathsf{Path})$

$\quad \mathsf{Elig} \leftarrow \{u \in \text{Out}(v) \mid u \notin \tau\}$ $\qquad \triangleright$ eligible (non-cycling) successors

$\quad$**if** $\text{Leaf}(v)$ **then**

$\quad\quad$**return** $v$

$\quad$**else if** $\mathsf{Elig} = \emptyset$ **then** $\qquad \triangleright$ deadend: no outgoing edges outside current trajectory

$\quad\quad$**if** $|\mathsf{Path}| = 1$ **then**

$\quad\quad\quad$**return** $v_0$ $\qquad \triangleright$ backtracked to root

$\quad\quad$**else**

$\quad\quad\quad u \leftarrow \text{pop}(\mathsf{Path}); \tau \leftarrow \tau \setminus \{u\}$ $\qquad \triangleright$ back up

$\quad\quad$**end if**

$\quad$**else**

$\quad\quad u^\star \leftarrow \underset{u \in \mathsf{Elig}}{\text{argmax}} \left( Q_{v,u} + c_{\text{puct}} P_{v,u} \frac{\sqrt{N_v}}{1 + N_{v,u}} \right)$

$\quad\quad \text{push}(\mathsf{Path}, u^\star); \tau \leftarrow \tau \cup \{u^\star\}$

$\quad$**end if**

**end while**

---

## A.4 EXPERIMENTAL SETUP FOR GAME24

**Hyperparameters**   The results in the main paper for ToT and HP are subject to hyperparameter search. In Table 4, we report the exact configurations that lead to the reported numbers.

Table 4: The used hyperparameter configurations for ToT and HP. HP is always run with verifications and shortcuts enabled (except for the ablation study). No hyperparameter search is conducted for the Unsolvable-100 dataset. Instead, the found setups in the ToT-Test-Split are used. Otherwise, very shallow search settings will dominate, simply due to them having fewer possibilities to make mistakes.

| Model | Dataset | Method | Parameters |
|---|---|---|---|
| Phi-4 | ToT-Test-Split | ToT | $b = 20, m = 11, \texttt{do\_verify} = \text{True}$ |
| | | HP | $k = 50$ |
| Mistral-Small-3 | ToT-Test-Split | ToT | $b = 20, m = 11, \texttt{do\_verify} = \text{True}$ |
| | | HP | $k = 200$ |
| Llama-3.3 (70B) | ToT-Test-Split | ToT | $b = 20, m = 11, \texttt{do\_verify} = \text{True}$ |
| | | HP | $k = 10$ |

**Infrastructure**   We run all experiments on compute nodes of four H100 (96GB) GPUs, 8 Intel Xeon Platinum 8468 CPUs, 1TB main memory running Rocky Linux. All runs (including the baselines) get one such node assigned until completion.

**Model serving**   LLMs were served via vLLM. The exact model identifiers on the Huggingface Hub are:

- `mistralai/Mistral-Small-24B-Instruct-2501`

- `microsoft/phi-4`

- `RedHatAI/Llama-3.3-70B-Instruct-FP8-dynamic`

In the last case we use quantized checkpoints to reduce memory footprint and to allow for a higher degree of parallelization. This way, all models fit on a single GPU which allows a data-parallel-size of four on our nodes. The chat completion requests are distributed in a round robin fashion to the separate workers, accumulated and returned to the application.

**Holistic Prompting Setup on Game24**   The search graph in Figure 3 shows an excerpt of two solved inputs and the boxes on the sides outline the prompts that were used. After finding a path to a solution node for input "(4 5 6 10)" by repeating steps (1a-5), the shortcut component in (1b) finds the appropriate action "(13 - 7 = 6)" to connect "(5 6 7 13)" to the winning path.

## A.5 DATASET STATISTICS

The ToT-Test-Split data (Table 6) is taken from the official repository (https://github.com/princeton-nlp/tree-of-thought-llm). The Unsolvable-100 dataset (Table 5) is constructed by solving all possible four digit instances of Game24, filtering for unsolvable combinations and sampling for a 100-element subset.

We report the complexity of the different datasets in Table 7 using collapsed, winning and minimal state/action counts. We further distinguish between local and global state collapse. In the first setting, we allow states to collapse within each problem instance, but not across multiple, which is instead captured in the global count. Finally, we report the number of all winning as well as the theoretical minimum states and actions required to solve each dataset completely. Minimum counts were obtained from the full global graphs by solving for the respective minimum Steiner Trees.

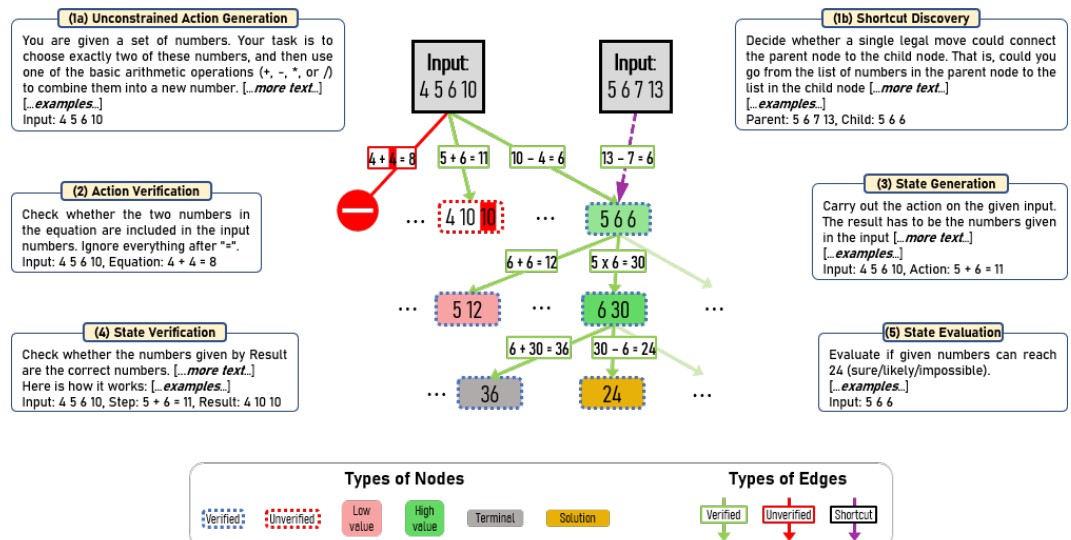

Figure 3: Holistic Prompting on a Game24 task

Table 5: All samples of the Unsolvable-100 dataset. Elements were drawn at random from the full set of unsolvable four-digit instances.

| | | | | |
|---|---|---|---|---|
| 1 6 6 7 | 2 9 11 12 | 1 6 10 11 | 3 3 4 10 | 7 7 7 9 |
| 1 1 2 5 | 6 9 11 11 | 5 7 11 12 | 4 7 7 10 | 3 3 5 11 |
| 3 7 10 12 | 3 10 10 11 | 1 2 8 11 | 6 6 7 8 | 7 7 8 12 |
| 3 3 11 11 | 1 1 1 4 | 4 4 9 9 | 2 3 5 12 | 1 7 11 11 |
| 2 10 10 10 | 1 10 10 10 | 5 6 7 11 | 1 2 8 12 | 4 7 7 12 |
| 1 8 9 9 | 9 10 10 10 | 2 5 5 6 | 2 3 9 11 | 7 11 11 12 |
| 1 4 11 11 | 5 5 5 10 | 5 5 7 12 | 2 2 9 9 | 8 8 11 11 |
| 8 10 10 10 | 8 9 10 11 | 1 1 5 11 | 1 4 11 12 | 1 5 7 7 |
| 1 2 10 10 | 1 9 10 10 | 1 1 2 4 | 4 11 12 12 | 5 5 5 11 |
| 6 6 10 11 | 2 9 9 9 | 4 11 11 11 | 1 7 10 11 | 1 10 11 11 |
| 1 1 5 10 | 5 5 5 7 | 1 6 6 8 | 5 5 7 9 | 5 8 9 9 |
| 1 1 5 9 | 9 10 11 11 | 5 8 9 10 | 8 9 9 9 | 5 8 10 10 |
| 1 3 7 11 | 9 9 10 11 | 1 8 11 11 | 4 9 9 11 | 1 1 4 11 |
| 2 9 9 10 | 5 8 8 11 | 1 1 6 10 | 1 1 5 12 | 7 8 8 8 |
| 3 3 7 10 | 1 4 7 10 | 6 7 7 7 | 4 5 5 12 | 1 1 1 10 |
| 7 9 9 12 | 5 6 6 11 | 1 6 7 7 | 7 7 8 8 | 1 9 10 11 |
| 1 1 3 3 | 5 5 6 9 | 6 7 7 12 | 1 2 9 10 | 2 6 7 7 |
| 2 5 11 11 | 3 5 9 11 | 1 1 7 11 | 3 6 7 11 | 6 10 10 12 |
| 9 10 10 12 | 1 1 7 7 | 2 5 5 5 | 5 6 8 11 | 9 9 11 11 |
| 6 6 9 9 | 7 7 7 8 | 9 11 11 12 | 1 6 10 10 | 1 5 5 7 |

## A.6 ERROR RATES

We analyze the types of errors that occur in the full reasoning graphs of Holistic Prompting on the ToT-Test-Split dataset, as reported in the main paper. We count four types of false positive errors and one general false negative error in Table 8.

Table 6: All samples of the ToT-Test-Split dataset, as curated by Yao et al. (2023a).

| | | | | |
|---|---|---|---|---|
| 4 5 6 10 | 1 5 9 13 | 3 4 9 13 | 2 6 8 13 | 3 4 6 6 |
| 1 2 4 7 | 5 6 7 13 | 4 5 10 12 | 8 8 10 12 | 5 8 8 8 |
| 2 5 8 11 | 5 5 8 10 | 1 2 7 11 | 1 3 8 13 | 6 8 8 12 |
| 3 4 4 13 | 2 4 6 12 | 4 5 6 8 | 4 4 7 10 | 2 3 4 9 |
| 6 7 8 9 | 6 7 8 11 | 6 10 12 13 | 1 7 10 13 | 2 6 7 11 |
| 1 11 11 13 | 7 9 9 13 | 1 3 9 9 | 1 9 10 13 | 5 9 12 12 |
| 1 8 10 11 | 3 6 9 12 | 1 4 4 11 | 3 3 4 11 | 1 2 7 12 |
| 2 3 6 9 | 6 9 12 13 | 2 3 9 10 | 2 5 7 7 | 2 4 5 6 |
| 1 3 5 9 | 4 7 9 13 | 1 2 3 13 | 3 9 10 13 | 5 5 8 13 |
| 3 3 7 12 | 5 6 8 12 | 1 6 6 6 | 2 3 4 7 | 2 3 3 10 |
| 4 5 7 9 | 2 4 6 7 | 1 2 2 9 | 4 4 8 12 | 3 4 8 12 |
| 1 2 8 13 | 2 5 10 10 | 1 3 6 11 | 1 2 6 10 | 2 4 6 11 |
| 4 6 6 9 | 6 6 7 12 | 5 10 12 13 | 1 5 12 12 | 2 2 8 9 |
| 1 4 4 8 | 6 9 9 11 | 2 3 6 6 | 5 6 6 8 | 1 5 6 7 |
| 1 5 10 11 | 5 8 11 12 | 6 7 10 12 | 7 7 8 11 | 5 8 10 11 |
| 3 4 6 11 | 5 6 8 10 | 7 8 8 12 | 1 3 7 10 | 4 4 9 12 |
| 2 4 8 9 | 6 11 12 13 | 3 4 6 8 | 3 3 9 12 | 2 5 6 6 |
| 1 4 5 13 | 2 2 8 8 | 1 7 9 11 | 3 5 7 10 | 2 4 9 12 |
| 2 2 7 12 | 2 7 12 13 | 2 3 6 13 | 4 10 12 13 | 4 8 11 13 |
| 3 3 6 7 | 2 6 8 12 | 2 2 5 12 | 2 3 10 12 | 4 9 10 13 |

Table 7: Dataset statistics for the Game24 task.

| Dataset | Unsolvable / total samples | States / Actions | | | |
|---|---|---|---|---|---|
| | | Locally collapsed | Globally collapsed | Winning | Minimal |
| ToT-Test-Split | 0 / 100 | 90K / 209K | 26K / 105K | 475 / 1232 | 48 / 148 |
| Unsolvable-100 | 100 / 100 | 61K / 136K | 20K / 75K | – | – |

Table 8: Summary of Error Types and counts of Holistic Prompting instances on ToT-Test-Split, as per Table 1. The high number of formatting errors on Phi-4 is mainly caused by its tendency to deviate from the formatting instructions. We frequently find it to switch to latex or python code to represent nodes and states. Some of these formats fell through our pipeline and thus might be incorrectly counted as errors. Interestingly, in the end, Phi-4 almost always manages to convert its intermediate steps to a solution in valid format and achieves excellent success rates.

| Error Type | Llama-3.3 | Phi-4 | Mistral-3 | Description |
|---|---|---|---|---|
| Math | 120 | 100 | 26 | The calculation was not carried out correctly. |
| Selection | 4 | 1 | 2 | Numbers outside given input used. |
| Apply | 0 | 0 | 10 | The new node was not formed correctly. |
| Formatting | 0 | 244 | 7 | The answer could not be parsed to an equation. |
| False Negative | 145 | 70 | 454 | A correct decision was flagged as incorrect. |

