# OpenReview forum: "Holistic Prompting: Joint Reasoning with Reusable States and Shortcut Discovery"
_ICLR.cc/2026/Conference — ICLR 2026 Conference Withdrawn Submission_

### Official Review · Reviewer_PWvx · 2025-10-27

**Soundness:** 2
**Presentation:** 1
**Contribution:** 2
**Rating:** 4
**Confidence:** 4

**Summary:**

This paper introduces Holistic Prompting (HP). The authors argue that conventional "trajectory-based state representations", where each state encodes its entire reasoning history, are redundant and prevent the reuse of intermediate computations, especially when tasks share overlapping subproblems. HP addresses this by processing multiple problem instances jointly within a shared And-Or graph structure, utilizing "collapsed states" that are Markovian and self-contained. This representation allows different reasoning paths to converge on and reuse identical subproblems, both within a single sample and across different instances. A core innovation of HP is an active "shortcut-discovery" mechanism, a type of inverse search that finds actions to connect existing unsolved subproblems to known, previously solved states, thereby aggressively pruning the search. Experiments demonstrate HP's effectiveness in the Game24 math puzzle and retrosynthetic planning.

**Strengths:**

1. The paper presents a novel idea on reusing reasoning states.
2. The proposed method is efficient in terms of tokens generated compared to ToT
3. The proposed method finds better performance over ToT
4. The shortcut discovery to intentionally arrive at already solved paths is interesting

**Weaknesses:**

1. The methodology seems to require common states that are exactly the same, so that different tasks could lead to common intermediate states, and the previous approach can be reused. Such tasks are rare, and the work only evaluates on 2 specific tasks.
2. The presentation is not clear. The descriptions are filled with jargon and not simple, intuitive explanations or illustrations of the underlying meaning.
3. The proposed methodology lacks a memory component. Thus, it is forced to solve all input problems simultaneously and could solve problems consecutively, storing intermediate results from prior steps.

**Questions:**

1. What other types of artificial or real-world problems can the proposed method solve?
2. Can a memory module be designed to store intermediate results for later reuse?

---

> ### Author Response · Authors · 2025-11-17
>
> Dear Reviewer,
>
> we appreciate your constructive feedback on our work and your service to the community.
> Concerning your questions:
>
> 1) `What other types of artificial or real-world problems can the proposed method solve?`
> We are currently in the process of adding two new tasks to broaden the scope of our work: boolean_expressions and multi_step_arithmetics, both taken from the 'big bench hard' benchmark.
>
> 2) `Can a memory module be designed to store intermediate results for later reuse?`
> Yes, and we already have support for this in our framework by controlling the 'width' ($b$) of the search (see, e.g., Algorithm 2).
> E.g., for $b=1$ we effectively solve one sample at a time (online processing) and for $b>1$ we get batch processing.
> However, you raise an import point: So far, we have not yet systematically explored the influence of this parameter, since all experiments were conducted on a fixed $b=10$.
> Thus, we ran an additional experiment and found $b$ to be a main driver of token efficiency:
> | $b$ | token usage |
> |---|---|
> | 1 | 10.3M |
> | 5 | 12.8M |
> | 10 | 13.1M |
> | 25 | 17.4M |
> | 50 | 17.6M |
> | 75 | 23M |
> | 100 | 22.4M |
>
> The success rates did not change significantly here.
> Thus, we could safely reduce token consumption by an additional 22%, however, at the cost of less parallelization.

---

### Official Review · Reviewer_k21N · 2025-10-28

**Soundness:** 2
**Presentation:** 3
**Contribution:** 2
**Rating:** 4
**Confidence:** 4

**Summary:**

This paper proposes Holistic Prompting, a prompting framework that enables Large Language Models (LLMs) to reuse intermediate reasoning results both within and across problem instances. Existing multi-step reasoning frameworks, such as Chain-of-Thought (CoT) and Tree-of-Thoughts (ToT), usually use trajectory-based state representations. Each state encodes the full reasoning history, preventing the reuse of partial reasoning outcomes and leading to redundant computation. To address this, Holistic Prompting constructs a shared state space of intermediate thoughts, supporting cross-instance reuse and shortcut discovery between solved and unsolved subproblems.
The proposed framework is empirically evaluated on two tasks (Game24 and retrosynthetic planning), showing improved success rates.

**Strengths:**

This paper introduces a unified framework for reasoning reuse and shortcut discovery, which conceptually bridges CoT/ToT-style prompting with retrieval-augmented reasoning paradigms.

**Weaknesses:**

- Limited Evaluation. The experiments focus on two specialized domains. For example, Game24 is a quite old synthetic dataset (used in ToT). As most results of the experiment and the Appendix are reported on this dataset, it remains a question whether the proposed method can be applied to more practical domains, such as tool-use tasks [3] and coding tasks [2].

- Comparison with Retrieval-Augmented Methods. The paper claims conceptual similarity to Retrieval-Augmented Generation (RAG) but does not include direct comparisons or ablations against RAG-based baselines that could also leverage reusable intermediate results [1].

[1]. Buffer of thoughts: thought augmented reasoning with large language models. NeurIPS 2024.

[2]. SWE-RL: Advancing LLM Reasoning via Reinforcement Learning on Open Software Evolution.

[3]. ToolRL: Reward is All Tool Learning Needs.

**Questions:**

The authors are encouraged to address the concerns above.

---

> ### Author Response · Authors · 2025-11-17
>
> Dear Reviewer,
>
> we appreciate your constructive feedback on our work and your service to the community. Concerning your remarks:
>
> 1) `Limited evaluation & scope`.
>
> Indeed, not all tasks that can be expressed in natural language have well-formed intermediate states. As such, it is currently unclear how our method would be impacted.
> To address this, we are working on adding a creative writing task to shed light on this issue. This type of task is particularly suited since different samples are almost guaranteed not to overlap.
> We are also implementing two additional tasks from 'big bench hard' (boolean_expressions, multi_step_arithmetics) to further broaden the scope of our work.
>
> 2) `Comparison with Retrieval-Augmented Methods is lacking.`
>
> Agreed. We ran additional experiments on the mentioned Buffer-of-Thoughts framework (Neurips Spotlight 2024), since they also study Game24 and show SOTA performance over graph-of-thoughts, meta-prompting, or PAL.
> Note that in their approach, they give models access to a python interpreter and let them write programs to derive the correct answer, which we do not consider for our own work.
>
> | Model | BoT (with tool use) | HP (ours) |
> |---|---|---|
> | Phi-4 | 78.9% | 96% |
> | Mistral 3 Small | 60% | 87% |
> | LLama 70B | 77.4% | 98% |
> | GPT-4 | 82.4 | |
>
> In terms of pure success rate, our method performs strictly better. In fact, our method enables even the smaller models (14B, 24B) to surpass GPT-4+BoT (and GPT-4+GoT, +PAL or +meta-prompting, for that matter).
> However, since BoT is a single-turn method and most of the heavy lifting is carried out by code execution, it is considerably cheaper in terms of completion tokens, often by a factor > 10.

---

### Official Review · Reviewer_wPXm · 2025-10-31

**Soundness:** 3
**Presentation:** 3
**Contribution:** 3
**Rating:** 2
**Confidence:** 4

**Summary:**

The authors focus on a "memoization" opportunity in LLM reasoning. Instead of making the LLM do its reasoning from scratch for each problem, they aim to discover and reuse common intermediate steps/results.

**Strengths:**

- The proposed approach connects unsolved problem instances to already-explored reasoning paths from other samples, which is a good contribution. It is similar to dynamic programming and can support more efficient decoding and token usage.
- The results show that while the success rates are similar, the required steps (in their domain captured as reaction and molecule nodes) are fewer.
- The error analysis and ablation results are good.

**Weaknesses:**

- The effectiveness of this approach depends on the presence of reusable sub-structures in the targeted task class; it is not clear how much overhead is introduced when overlap is minimal (or non-trivial to adapt).
- Aggressive pruning, while controlling complexity, risks prematurely discarding valuable reasoning paths for atypical instances, potentially missing correct or novel solutions. The authors should think of situations where this can happen.
- There is a general assumption of using the LLM for batch or clustered problem solving rather than one-shot, highly individualized queries, potentially limiting its applicability in interactive or open-ended settings. (This is fine, but it needs to be acknowledged and mentioned).
- It would have been ideal if the authors could have connected this to RAG architectures and talked about situations where sub-structures are re-used even across problem settings or domains.

**Questions:**

- Please address the points raised in the weaknesses section.

---

> ### Author Response · Authors · 2025-11-18
>
> Dear Reviewer,
>
> we thank you for the constructive feedback and the service to the community. Regarding your concerns:
>
> 1) `The effectiveness of this approach depends on the presence of reusable sub-structures in the targeted task class; it is not clear how much overhead is introduced when overlap is minimal (or non-trivial to adapt).`
>
> In the case of non-overlapping samples our framework can, in principle, perform non-necessary queries to find matches between states. We argue that this can be effectively mitigated with an appropriate shortcut evaluator and shortcut limit ($k$) (cf. Algorithm 2).
> To assess this, we are working on a creative writing task where intermediate steps are rarely overlapping.
>
> 2) `Aggressive pruning, while controlling complexity, risks prematurely discarding valuable reasoning paths for atypical instances, potentially missing correct or novel solutions. The authors should think of situations where this can happen.`
>
> Pruning, in our work, refers to stopping the search on a specific problem instance as soon as the first solution to it is found.
> We may not exhaust all possible solutions to a sample, although it would be straightforward to optimize for it.
> We also never discard paths that have not been exhausted yet. Whether a node is exhausted depends on an LLMs ability to find ways to expand from there. Consequently, any LLM based search (including ToT, GoT, and ours) immediately loses probabilistic completeness if the LLM demonstrates zero-probability of visiting some state.
> We will reflect the possibility of probabilistic incompleteness in the manuscript.
>
> 3) `There is a general assumption of using the LLM for batch or clustered problem solving rather than one-shot, highly individualized queries, potentially limiting its applicability in interactive or open-ended settings. (This is fine, but it needs to be acknowledged and mentioned).`
>
> We agree that our manuscript is currently focused on tasks with specific properties (overlapping substructures, batch solving). We will emphasize the restrictions and what follows from them (limited interactive or open-ended applicability) and thank the reviewer for raising the issue.
>
> Concerning online vs. batch processing, we will also include a new ablation study, where we found that reuse occurs more frequently on smaller batches and most frequently in online processing. Below is an overview over different batch sizes ($b$) and the associated token consumption (Phi-4, Game24):
>
> | $b$ | token usage |
> |---|---|
> | 1 | 10.3M |
> | 5 | 12.8M |
> | 10 | 13.1M |
> | 25 | 17.4M |
> | 50 | 17.6M |
> | 75 | 23M |
> | 100 | 22.4M |
>
> Note that the success rate stayed at ~98% in all settings. We see a trade-off between parallelizability and reuse, caused by solving states in each batch in isolation and thereby limiting reuse. Future work could therefore look into selecting more diverse batches that are less likely to converge on the same outcome.
>
> 4) `It would have been ideal if the authors could have connected this to RAG architectures and talked about situations where sub-structures are re-used even across problem settings or domains.`
>
> We believe this to be an issue of context compatibility. Whether and how a part of one approach qualifies for reuse in another setting may depend on structure (e.g., types, constraints, units, symmetries) and representations/schemas (e.g., variable names, isomorphisms, surface forms). Connecting this to RAG, we understand our proposed graph structure as a knowledge base of reusable components that is queried by our shortcut component, which essentially acts as a retriever. We will make this clearer in the manuscript.
> For future work, we envision a system in which the retriever searches for compatible reasoning steps not only based on semantic or surface similarity but also on structural and logical equivalences to match contexts.
>
> We also added a new RAG baseline (as proposed by Reviewer k21N). The Buffer of Thoughts (Neurips Spotlight 2024) utilizes retrieval of previous thoughts and consequent thought distillation to achieve SOTA performance over Graph of Thoughts, Meta-Prompting and others. They also use the same Game24 task. Brief results are below:
>
> | Model | BoT (with tool use) | HP (ours) |
> |---|---|---|
> | Phi-4 | 78.9% | 96% |
> | Mistral 3 Small | 60% | 87% |
> | LLama 70B | 77.4% | 98% |
> | GPT-4 | 82.4 | |
>
> In terms of success rate, our work is significantly stronger and allows our smallest model (Phi4, 14B) to surpass GPT-4 even when equipped with BoT, GoT, and others.
> However, BoT heavily leverages tool-use and gives LLMs access to a python interpreter to run generated code, which we do not consider. By offloading much of the required computation, BoT generally requires far fewer output tokens (often by a factor > 10), although this does not suffice to achieve higher success rates.

---

### Official Review · Reviewer_rbqF · 2025-11-07

**Soundness:** 2
**Presentation:** 2
**Contribution:** 2
**Rating:** 4
**Confidence:** 4

**Summary:**

This paper introduces a new reasoning method for LLMs as an alternative to chain of thought (CoT) and tree of thought (ToT). The novel contribution is to build a graph of thoughts which is shared across input samples, where edges are built between intermediate reasoning states such that the states can be reused across different samples, thereby cutting down the length of reasoning traces. The states can only be reused as exact matches (as opposed to clusters or other abstractions). Two experiments are presented, the first a simple arithmetic problem using LLMs as the base predictor and the second a chemical synthesis problem, where existing domain-specific predictors were used instead of LLMs, due to LLMs giving high errors in this domain. The results on the arithmetic problem noticeably outperformed CoT and ToT with significantly fewer intermediate states and model calls. In the chemistry task, it matched the already high performance of the existing baselines but with fewer intermediate states.


The paper seeks to tackle an important problem and the idea of reusing intermediate reasoning states across inputs is definitely promising. However, in its current form, I am not convinced that this method will allow such an architecture to actually scale to standard LLM text-based reasoning problems, due to a combination of my intuition about the architecture and the lack of results on complex text-based domains. If the equality test was abstracted into some form of clustering or high-level concept correspondence, that may be a different story as this could potentially compress complex state spaces. For now, I don’t believe the method is competitive.

**Strengths:**

* The paper is clearly written
* It tackles the important and well-motivated problem of intermediate state representation and reuse in LLM reasoning
* The main methodological contribution, a sample-shared graph allowing reuse seems novel, though bear in mind that some very recent (last couple of months) methods tackle the reuse problem (e.g. metacognitive reuse, cross-question method reuse)

**Weaknesses:**

* The scope seems very limited. Since the matched intermediate states must be very/exactly similar and are low-level states without any abstraction, it is hard to see how this method could extend beyond problems with very simple input and intermediate token sequences. If the intermediate states were whole paragraphs or even sentences, how could these be reused at all?
* This paper is presented as a method for reasoning over LLMs, yet the second experiment didn’t use LLMs at all. If the problem precludes LLMs, you might as well use a domain-specific method rather than reasoning.
* I’m not sure why Game24 was tested with simple baselines, without considering higher performing ones like Graph of Thoughts or Self-discover, which may also be more relevant methodologically. It’s reasonably likely these methods would have matched your performance.

**Questions:**

See Weakness section.

---

> ### Author Response · Authors · 2025-11-17
>
> Dear Reviewer,
>
> Thank you for your constructive feedback on our work and for your service to the community. Regarding your remarks:
>
> 1) `The scope seems very limited. Since the matched intermediate states must be very/exactly similar and are low-level states without any abstraction, it is hard to see how this method could extend beyond problems with very simple input and intermediate token sequences. If the intermediate states were whole paragraphs or even sentences, how could these be reused at all?`
>
> We agree that our current matching strategy naturally limits the tasks that can be adapted for reuse. In the original submission, we acknowledged this up front (e.g., in the abstract) and discussed the constraint (e.g., Section 7), but clearly did not emphasize it enough.
>
> We plan to extend our experiments along two dimensions:
> * We will perform an ablation study on a task whose intermediate states are full paragraphs (e.g., a creative-writing setup) to quantify how reduced state reusability impacts our method’s performance.
> * We will add two problems from BIG-Bench Hard (boolean-expression, and multi-step arithmetic) to demonstrate that our framework has applications in relevant contemporary benchmarks.
>
> 2) `This paper is presented as a method for reasoning over LLMs, yet the second experiment didn’t use LLMs at all. If the problem precludes LLMs, you might as well use a domain-specific method rather than reasoning.`
>
> We appreciate the reviewer’s point and agree that Experiment 2 does not instantiate the action/value oracles with an LLM. This was a deliberate choice to isolate the contribution of our framework from the quality of the single-step generator in retrosynthesis. Current chemistry LLMs are not yet reliable enough to make a meaningful evaluation of a multi-step planner. As reported by prior work (Guo et al., Neurips), single-step disconnection error rates for GPT-4 remain over 88.6%, even with scaffolding. In our setup, neither general-purpose nor chemistry LLMs solved a single target in our test set. Under such conditions, the end-to-end outcome is dominated by uninformative single-step failures rather than by the planner, making it impossible to assess the value of the reasoning layer.
> Note that our framework already supports LLMs, since we interface with action/value generation as black boxes. Therefore, we position the retrosynthesis task as a durable test bed for multi-step reasoning.
>
> 3) `I’m not sure why Game24 was tested with simple baselines, without considering higher performing ones like Graph of Thoughts or Self-discover, which may also be more relevant methodologically. It’s reasonably likely these methods would have matched your performance.`
>
> We ran additional experiments on the Buffer of Thoughts framework (Neurips Spotlight 2024), since they also study Game24 and show SOTA performance over Graph of Thoughts, Meta-prompting, or PAL.
> Note that in their approach, they give models access to a python interpreter and let them write programs to derive the correct answer, which we do not consider for our own work.
>
> | Model | BoT (with tool use) | HP (ours) |
> |---|---|---|
> | Phi-4 | 78.9% | 96% |
> | Mistral 3 Small | 60% | 87% |
> | LLama 70B | 77.4% | 98% |
> | GPT-4 | 82.4% | |
>
> Our method considerably outperforms BoT (and by extension: Graph of Thought, Meta Prompting, PAL) in terms of success rate.
> Even the smallest LLM in our experiment (Phi-4, 14B) now beats GPT-4.
> However, since BoT is a single-turn method and most of the heavy lifting is carried out by code execution, it is considerably cheaper in terms of completion tokens, often by a factor > 10.
> Investigating the extent to which our framework can also benefit from tool use is an important direction for our future work.

---

### Note · Authors · 2026-01-16

I have read and agree with the venue's withdrawal policy on behalf of myself and my co-authors.